# Peer review of "Modification of Type B Inclusions by Calcium Treatment in High-Carbon Hard-Wire Steel"

_metals, doi:10.3390/met11050676_

Round 1

Reviewer 1 Report

Dear Authors,

Congratulations on the high scientific level of the article. I encourage you to do more research in this area.  I accept the article for publication in its current form.

Best regards,

Author Response

Reply to the reviewer: Thank you for your suggestion.

Reviewer 2 Report

The paper presents a study on modification of type B inclusions by calcium treatment in high-carbon hard-wire steel. In order to be published it needs changes.
(1) Firstly, the English, mainly of the abstract, has to be revised. For example, the first sentence is too long which makes reading and understanding difficult. In addition, in the second paragraph there are capital letters in the middle of the text. Authors should avoid starting a sentence with "it".
(2) Authors must add a space between the number and their unit throughout the document.
(3) In the "Experimental method" section there is a lack of information, namely the EDS technique and also the procedure for determining the size of inclusions.
(4) In my opinion it is necessary to add a global microstructure of the steel to later evaluate and go into detail about the inclusions, at the beginning of section 3.1.
(5) Figure 2 needs to be better explained in the text. Furthermore, it is not clear why the authors use maps of different elements for each type of inclusion. They should also avoid presenting samples with just the number. Please present the conditions / composition instead of "sample 1, sample 2, ...)
(6) Sections 3.2 and 3.3 have the same title with only the words in different order. There is no point in having two sections on the same subject.

Author Response

(1).Reply to reviewer: Thank you for your suggestion, we have checked and revised the manuscript, hoping to achieve better presentation effect. 

(2).Reply to reviewer: We have added a space between the number and units in the manuscript. The amendments are as follows: (18), (57), (61), (63), (76-78), (86-87),(129-134),(158-160),(183).

(3).Reply to reviewer: In the experimental method part, the method of inclusion statistics is added in the experimental method part. The amendment is: (89, 94-96)

(4).Reply to reviewer: Inclusions with low deformation rate induce cracks in high carbon hard wire steel during drawing. Mainly due to the different thermal expansion coefficients between inclusions and steel matrix, a radial tensile force is generated in the matrix around inclusions, which leads to stress concentration around inclusions. When inclusions change to spherical shape, the stress concentration around inclusions weakens, which improves the drawing performance of high carbon hard wire steel. The amendment is: (99-104)

Because the main content in this paper is the modification of inclusions and characteristics of inclusions, but not the relationship between the effect of inclusion on the microstructure of steel, we think it is not necessary to illustrate the microstructure of steel.

(5).Reply to reviewer: According to your suggestion, the composition of experimental steel is added in the title of the drawing. Figure 2 is used to illustrate the elements of Ca and Al distributed in inclusions homogeneously. The amendment is: (114-115; 121-122)

(6).Reply to reviewer: Section 3.2 describes the size distribution of inclusions and section 3.3 describes the areal density distribution of inclusions. We have changed the title of Section 3.3. The amendment is: (174)

Reviewer 3 Report

Experiments and thermodynamic calculations were performed for alumina inclusion modificaiton by calcium treatment for high carbon steel for wire rods.
Novelty is recognized in that the target steel is a high carbon steel.
However, a clearer necessity for the research purpose is required to be mentioned.
In-depth consideration is required, including comparison between high carbon concentration steel and low carbon concentration steel.

Author Response

Reply to reviewer: According to your suggestion, Thank you for your suggestion. We will discuss the difference between high carbon steel and low carbon steel in the later research.

Round 2

Reviewer 2 Report

The paper can be now accepted for publication. 

Author Response

Reply to the reviewer: Thank you for your suggestion